# Determination of Occurrences, Distribution, Health Impacts of Organochlorine Pesticides in Soils of Central China

**DOI:** 10.3390/ijerph16010146

**Published:** 2019-01-07

**Authors:** Tekleweini Gereslassie, Ababo Workineh, Onyango Janet Atieno, Jun Wang

**Affiliations:** 1Key Laboratory of Aquatic Botany and Watershed Ecology, Wuhan Botanical Garden, Chinese Academy of Sciences, Wuhan 430074, China; tekle206@gmail.com (T.G.); abiyework@gmail.com (A.W.); janetonyango74@gmail.com (O.J.A.); 2Sino-Africa Joint Research Center, Chinese Academy of Sciences, Wuhan 430074, China; 3Department of Pollution Ecology, University of Chinese Academy of Sciences, Beijing 100049, China

**Keywords:** organochlorine pesticides, dichlorodiphenyltrichloroethane, hexachlorocyclohexanes, cancer risk, land-use

## Abstract

Organochlorine pesticides are groups of chemicals applied to prevent pest and insect infestation. This study was aimed at investigating the concentration, potential sources, cancer risk and ecological toxicity of organochlorine pesticides (OCPs) in Huangpi district, Wuhan, China. Eight OCPs in soil samples collected from four land-use types at depths of 0–10 and 10–20 cm were examined. Sample extraction was carried out by solid phase matrix extraction method and analyzed using Agilent gas chromatograph 7890B equipped with electron capture detectors (ECD). The total concentration of OCPs ranged from 0.00–32.7 ng g^−1^ in the surface and 0.01–100.45 ng g^−1^ in the subsurface soil layer. Beta hexachlorocyclohexanes (β-HCH) with 2.20 and 7.71 ng g^−1^ in the surface and subsurface soil layers, respectively, was the dominant compound. The mean concentrations of OCPs in all samples were less than the threshold values for hexachlorocyclohexanes (HCHs) and dichlorodiphenyltrichloroethane (DDTs) in China soil. Concentration of OCPs in the four land-use types were in the order of: paddy field > barren land > farmland > plastic greenhouse. Results of composition analysis revealed recent application of lindane as a major and historical use of new technical HCHs as a minor source of HCHs. On the other hand, application of new technical p,p’-DDT is the main source of DDTs in the study area. The estimated lifetime average daily dose, incremental lifetime cancer risks and hazard quotient values revealed that there is less likelihood of carcinogenic and noncarcinogenic health risks on the local residents.

## 1. Introduction

Organochlorine pesticides (OCPs) are groups of chemicals applied to prevent and eradicate insects, weeds, fungi and bacterial impacts [1]. Dichlorodiphenyltrichloroethane (DDTs) and hexachlorocyclohexanes (HCHs) are among the extensively used OCPs in agriculture, public health, in controlling termite, tsetse fly and malaria causing mosquitoes [2,3,4,5]. Organochlorine pesticides are known for their ecological toxicity, high persistence and strong affinity to bioaccumulation [6,7]. Due to the high level of environmental toxicity and health impacts, the large-scale production and use of OCPs (HCHs and DDTs) is currently prohibited. However, the environmental, human and wildlife health impacts of their residues have been detected and reported in many countries [8,9,10]. In addition to their persistence, the continuous production, illegal distribution and use of HCHs and DDTs as agricultural inputs has been apparently reported in many countries, particularly in developing countries [11]. The developing and underdeveloped countries alone use 4000–5000 tons of DDTs annually for vector control applications [12].

Organochlorine pesticides originated from industrial discharges, domestic/urban wastes, agricultural application, and other point sources and nonpoint sources enter into geological components through runoff, air and other agents [7]. Due to their wide range of past applications, human health and environmental impacts, HCHs and DDTs are often considered as representative compounds in evaluating the environmental status of OCPs [13]. Both HCHs and DDTs have a high possibility of long-range transportation, explained as grasshopper effect [1,12], which might result in moderate, high and acute chronic health impacts including those of non-producer and non-user countries [1,14].

Extensive production and utilization of OCPs, particularly HCHs and DDTs for agriculture and other purposes, has been reported in China [14,15]. China was the leading OCP producer and consumer country until the 1990s [16,17], producing approximately 4.99 × 10^6^ tons of HCHs and 0.49 × 10^6^ tons of DDT in 30 years [17,18,19]. DDTs and HCHs contribute more than three quarters of the OCPs in China [20] and are often considered as the primary agents for many chronic diseases such as cancer, endocrine disruption, and mutagenesis [21]. According to research, the application of technical DDTs, technical HCHs and lindane take the lion’s share role in the buildup of China’s environmental pollution [14,22]. Thus, DDTs and HCHs are among the most researched ubiquitous organic compounds in China [18]. Pollution from DDTs and HCHs have been reported in different environments, mainly aquatic bodies of China, particularly for Wuhan, in Yangtze River [23] and East Lake [24]. To our knowledge, there is a dearth of research works conducted on the status and distribution of OCPs in agricultural soils of Wuhan and its vicinities.

Rapid industrialization, urbanization and agricultural inputs play a big role in increasing environmental concentration and distribution of OCPs in Wuhan [24]. Huangpi is one of the urban districts located in the north of Wuhan city. Construction sites, forestry, farming, and barren lands are the predominant land-use types in the district. It has a large area for crop production and other related agricultural practices [25]. The study site is known for its potential productivity and intensive agricultural practices, which might require a significant amount of agricultural inputs. Producers in the study area have a strong habit of using fertilizers, herbicides, pesticides, and fungicides to increase the potential productivity and ensure food self-sufficiency. Besides, local residents use DDTs to prevent malaria and mosquito infestation. Both local and national governments are encouraging intensive agricultural production to feed the continuously increasing population in China in general and the surrounding communities in particular [26,27]. The dietary requirement for large portion of the population in the city of Wuhan and Huangpi district depends on the crops, vegetables and fruits produced from the district. As a result, human exposure to DDTs and HCHs through direct consumption, dermal contact and inhalation is expected. Thus, studying the current status and potential impacts of OCPs is vital. Therefore, the aim of this study was to examine the concentration, potential sources and ecological and human health risks of DDTs and HCHs in Huangpi soils from four land-use types.

## 2. Materials and Methods

### 2.1. Sample Collection and Pretreatment

Soil samples were collected from 18 sampling places located in Huangpi district, Wuhan, China. The study site is located between 30°52′30″ N and 114°22′30″ E. The district covers 56,700 hectares of farmlands, hosting a total population of 874,938 [28]. Samples were collected from four different land-use types, namely barren lands (BL), farmland (FL), plastic greenhouse (PGH) and paddy field (PF), at depths of 0–10 and 10–20 cm. Samples were collected using a precleaned stainless steel grab sampler. Locations for each sampling points were recorded using a global positioning system (GPS). Samples 17 and 18 were collected from a similar location, thus one GPS reading was recorded for these two samples. Thus, there were 17 total sampling sites (Figure 1).

Three replications with a 250 g wet weight were collected and immediately wrapped in polyethylene ziplock bags [14]. The three replications were mixed and lyophilized using a bench top lab vacuum freeze dryer at a −40 °C for 48 h. All samples were ground and sieved through a 100 mesh (0.149 mm) stainless steel sieve [29] and stored in a refrigerator at a −20 °C until the next extraction steps [30].

### 2.2. Chemicals and Standards

The OCPs reference standards of 1000 mg L^−1^ containing α-HCH, β-HCH, γ-HCH, δ-HCH, p,p’-DDE, p,p’-DDD, p,p’-DDT and o,p’-DDT was purchased from AccuStandard Inc. (AccuStandard, New Haven, CT, USA). Working standards of OCPs was prepared by diluting the stock solution in n-hexane. Instrumental calibration was carried out using calibration standard solutions of 0.1, 0.2, 0.5, 1, 2, and 10 μg L^−1^. The graphs for the calibration were linear with the average correlation coefficient value (R^2^ = 0.999). A Chromatographic grade dichloromethane (Fisher Scientific, Waltham, MA, USA) and acetonitrile (Mallinckrodt Baker, Inc., Phillipsburg, NJ, USA) and n-hexane was purchased from (Sinopharm Chemical Reagent Co., Ltd., Shanghai, China). In addition, analytical grade reagents, namely carbon (C18) (SiliCycle, Inc., Quebec City, QC, Canada), anhydrous sodium sulfate (Na_2_SO_4_) and copper powder (Cu) (Sinopharm Chemical Reagent Co., Ltd., Shanghai, China), Florisil (Beijing Yizhong Chemical Plant, Beijing, China) and neutral silica gel (Qingdao Haiyang Chemical Co., Qingdao, China), were purchased. Pretreatment and activation of the reagents were carried out to improve the effectiveness of the extraction. Accordingly, the anhydrous sodium sulfate was baked at 450 °C for 4 h before use. The 60–100 mesh Florisil and 100–200 mesh neutral silica gel were activated in a dry oven at 150 °C for 10 h and 180 °C for 4 h, respectively [24]. The neutral silica gel was deactivated with 3% ultra-pure water before use. Unlike other reagents, copper was activated by soaking the copper powder in a 2 N of HCl (Sinopharm Chemical Reagent Co., Ltd., Shanghai, China) for 12 h at room temperature, and washed three times with water and three times with acetone (Mallinckrodt Baker, Inc., Phillipsburg, NJ, USA).

### 2.3. Sample Extraction, Cleanup and Analysis

The target compounds were extracted using a modified matrix solid-phase dispersion extraction method [3]. One gram of soil sample was thoroughly blended and homogenized with three grams of C18 using a glass mortar and pestle for 5 min [29]. One gram of anhydrous sodium sulfate (Na_2_SO_4_), 1 g of Florisil, and 1 g of neutral silica gel were added to remove any trace of water and form a free-flowing powder [31] and then 1 g of activated copper powder was added to remove elemental sulfur [24,31]. The activated Na_2_SO_4_, Florisil, neutral silica gel, and copper powder were packed into a 10 mL syringe barrel column with 0.22 μm membrane filter from bottom to top. After the blended mixture had completely transferred into the syringe column using a funnel, the column was closed with 0.22 μm membrane filter and compressed with the syringe plunger to remove air from the syringe. Final elution was carried out using 20 mL of dichloromethane by gravity flow; before the extracts have concentrated till dryness under gentle high purity nitrogen (N) stream. Finally, the dried residue was re-dissolved in 100 μL of n-hexane [24].

### 2.4. Instrumental Analysis

An Agilent gas chromatograph 7890B electron capture detectors (ECD) was used to determine the levels of DDTs and HCHs in the soil samples. One microliter of sample extracts was automatically injected into DB–1701 (30 m × 320 μm × 0.25 μm) capillary column with a runtime of 30 min [5]. The gas chromatograph column temperature was programmed to begin at 120 °C (held for 1 min) and ramped to 240 °C at a rate of 7 °C/min and held for 0.5 min, whereas the injector and detector were operated at 270 and 300 °C, respectively. To ensure the experimental quality, all glassware was thoroughly cleaned with distilled water, baked in a muffle furnace (Zhengzhou Protech Furnace Co., Ltd., Zhengzhou, China) at 450 °C for 4 h and rinsed with hexane before use. Interference and cross contamination between samples were estimated by including a procedural blank for each set of ten samples [24]. No OCPs were detected in the blank samples and the average recoveries in this study ranged 75–110%. The detection limits (DL) and quantification limits (QL) for soil samples varied within 0.046–0.050 ng g^−1^ and 0.159–0.165 ng g^−1^, respectively.

In addition to OCPs, three selected soil properties were determined by following the methods used by [32]. The total organic carbon (TOC) in the soil was analyzed with the Shimadzu TOC analyzer (Shimadzu, Hadano Kanagawa, Japan) (SSM–5000A). Soil pH was determined using a pH meter in a ratio of 1:5 soil: water [33,34]. Soil moisture content was determined by drying the known weight of soil in a dry oven at 105 °C for 24 h.

### 2.5. Human Health and Cancer Risk Assessment

Human exposure to DDTs and HCHs can induce serious health effects such as endocrine disruption, cancer, immunologic, and neurological problems [35]. The carcinogenic and noncarcinogenic human health impacts of OCPs may vary with age, habits, health conditions and rate of exposure to harmful chemicals [5]. The human health impact of OCP is commonly evaluated using Lifetime Average Daily Dose (LADD), Incremental Lifetime Cancer Risks (ILCR) and Hazard Quotient (HQ) [36]. ILCR value ≤ 10^−6^ represents very low risk, 10^−6^ ≤ value ≤ 10^−4^ low risk, 10^−4^ ≤ value ≤ 10^−3^ moderate, 10^−3^ ≤ value ≤ 10^−1^ high and value ≥ 10^−1^ very high cancer risk [5,21,36,37]. Likewise, HQ < 1 and LADD ≤ 10^−6^ for OCPs in soil represent an acceptable risk level [38]. According to the literature, cancer risk and other health impacts of OCPs occur mainly through three exposure pathways: breathing, direct skin contact and consumption [39]. The LADD, HQ and ILCR through consumption, dermal contact and breathing were estimated using Equations (1)–(6) adopted from [36,39].
(1)$$\mathrm{LADD}=\frac{C_{\mathrm{soil}}\times \mathrm{IngR}\times \mathrm{EF}\times \mathrm{ED}\times \mathrm{CF}}{\mathrm{BW}\times \mathrm{AT}}$$
(2)$$\mathrm{ILCR}=\frac{\mathrm{LADD}}{\mathrm{CSF}}$$
(3)$$\mathrm{HQ}=\frac{\mathrm{LADD}}{\mathrm{RFD}}$$
(4)$${\mathrm{CR}}_{\mathrm{ingest}}=\frac{C_{\mathrm{soil}}\times \mathrm{IngR}\times \mathrm{EF}\times \mathrm{ED}\times \mathrm{CF}\times {\mathrm{SF}}_{\mathrm{oral}}}{\mathrm{BW}\times \mathrm{AT}}$$
(5)$${\mathrm{CR}}_{\mathrm{dermal}}=\frac{C_{\mathrm{soil}}\times \mathrm{SA}\times {\mathrm{AF}}_{\mathrm{soil}}\times \mathrm{ABS}\times \mathrm{EF}\times \mathrm{ED}\times \mathrm{CF}\times {\mathrm{SF}}_{\mathrm{oral}}\times \mathrm{GIABS}}{\mathrm{BW}\times \mathrm{AT}}$$
(6)$${\mathrm{CR}}_{\mathrm{inhale}}=\frac{C_{\mathrm{soil}}\times \mathrm{InhR}\times \mathrm{EF}\times \mathrm{ED}\times \mathrm{IUR}\times {\mathrm{AF}}_{\mathrm{Inh}}}{\mathrm{PEF}\times \mathrm{AT}}$$
where C_soil_ is the concentration of the OCPs in soil (mg kg^−1^), IngR is the soil ingestion rate (100 mg days^−1^), ED is the exposure duration (70 years for adult and 12 years for children), EF is the assumed exposure frequency (365 days/year), CF is the conversion factor (1 × 10^−6^ kg mg^−1^), AT is the upper–bound value of averaging time (70 × 365 = 25,550 days for adults, 4380 days for children), BW is the average body weight (70 kg adults and 27 kg for children), SA is the contact surface area of skin with soil (3300 cm^2^), AFsoil is the Skin adherence factor for soil (0.2 mg cm^2^), ABS is the dermal absorption factor% (0.2 for DDTs and 0.1 for HCHs), GIABS is the fraction of contaminant absorbed in gastrointestinal tract (1), AFInh is the absorption factor for the lungs (1), InhR is the inhalation rate (15.8 m^3^ days^−1^ for adults), SForal is the oral slope factor (0.2 mg kg^−1^ days^−1^), PEF is the particle emission factor (1.36 × 10^−9^ m^3^ kg^−1^), IUR is the inhalation unit risk (0.057 mg m^3^), and CSF is the cancer slope factor (0.007 mg kg^−1^ days^−1^). CR_ingest_, CR_dermal_, and CR_inhale_ are cancer risk via ingestion, dermal contact and inhalation of soil, respectively. RfD is the reference dose (2 mg kg^−1^ days^−1^). All the values of selected parameters adopted from health Canada federal contaminated site risk assessment in Canada and [38].

### 2.6. Data Analysis

All descriptive statistics ranges, mean and standard deviation were calculated using IBM SPSS statistics version 20. One-way analysis of variance (ANOVA) was applied to analyze the statistical differences in the mean concentrations OCPs with soil depth and individual OCPs at a significance of α < 0.05. Pearson’s correlation analysis was also used to determine the connection between PAHs and soil properties, using a two-tailed test (α = 0.05 and 0.01).

## 3. Results

### 3.1. Concentrations of Organochlorine Pesticides in Huangpi Soils

The range, mean, standard deviation, frequency, DL and QL for HCHs and DDTs investigated in the present study are presented below (Table 1). The detection frequency of OCPs in this study ranged 39–94% (0–10 cm) and 56–100% (10–20 cm). The levels of ∑OCPs (sum of the HCHs and DDTs) in the subsurface soils ranged 0.01–100.45 ng g^−1^ and were higher than the ND–32.7 ng g^−1^ in the surface soils (Table 1). Similarly, a slightly lower concentration of OCPs in the surface layer than the subsurface was reported from ditch wetlands of Chinese estuaries [6]. The ∑OCPs in this study were relatively lower than those of reported from soils of North Pacific Ocean (5.14–676 ng g^−1^) [21] and sediments of Weihe River, China (291.16 ng g^−1^) [14]. It was however relatively higher than those of from surface water of central China (0.004–0.011 ng g^−1^) [15] and a mean concentration of OCPs in Xinghua Bay soil, China (2.75 ng g^−1^) [40].

The mean concentration of HCHs in Huangpi soil were in a descending order of β-HCH > δ-HCH > α-HCH > γ-HCH in the surface, and β-HCH > γ-HCH > δ-HCH > α-HCH in the subsurface soil layers. β-HCH was the abundant and evenly distributed at both soil depths. The abundance of β-HCH is due to its less degradation, water solubility, and high affinity to adsorb in soil [41,42] as well as the conversion of γ-HCH isomer to β-HCH by microorganisms and photoisomerization [15,43]. A similar phenomenon is reported in soils from Bohai Sea, China [2] and soils from Lahore city, Pakistan [44]. The average concentration ranges of HCHs in this study (Table 1) were similar to those reported in soils of Xinghua Bay (1.22–7.47 ng g^−1^) [40], soils along Bohai Sea (3.5 ng g^−1^) [2] and from sediments of CauBay River, Vietnam (7.82 ng g^−1^) [39]. Meanwhile, they were noticeably higher than those of Wolong natural reserve soils (0.15–1.35 ng g^−1^) [45]. The one-way analysis of variance (ANOVA) showed that there was no a statistically significant variation in the mean concentration of individual OCPs at the two soil depths (F(1,14) = 0.585, *p* = 0.457) and between mean values HCHs and DDTs across all sampling points (F(1,34) = 2.732, *p* = 0.108) at α < 0.05.

The concentration level of DDTs in the present study were in descending order of p,p’-DDT > o,p’-DDT > p,p’-DDE > p,p’-DDD (0–10 cm) and p,p’-DDT > p,p’-DDE > o,p’-DDT > p,p’-DDD (10–20 cm). The obtained concentration of DDTs in the current study was lower than the concentration in soils of Xinghua Bay (0.91–27.89 ng g^−1^) [40], sediments of CauBay River, Vietnam (68.35 ± 13.57 ng g^−1^) [39], soils along with Bohai Sea, China (17 ng g^−1^) [2] and soils from Beijing, China (38.66 ng g^−1^) [46]. The comparison concentration of OCPs from different works of literatures is summarized in Table 2.

### 3.2. Distribution of Organochlorine Pesticides in Huangpi across Land-Use Types

The sum of mean concentrations of OCPs (HCHs and DDTs) from four land-use types across the study site from two depths (0–10 cm and 10–20 cm) are displayed in Table 3 and Figure 2 and Figure 3. The sum of concentrations of OCPs in the eighteen soil samples varied within 0.091–56.38 ng g^−1^ (0–10 cm) and 0.023–155.93 ng g^−1^ (10–20 cm). The concentration of OCPs also showed small differences among the four land-use types sampled in the present study. The concentrations of OCPs across the four land-use types and individual samples are summarized in Table 3.

According to the literature, variation in OCPs levels among agricultural soils highly depends on the existing and previous land-use types [6]. As illustrated in Table 3, the sum of mean concentration of OCPs found in the surface layers of the four land-use types were in a sequential order of: PF > BL > FL > PGH. However, the level of OCPs in the subsurface layer was higher in PF followed by FL, PGH, and BL, respectively. The pooled arithmetic mean concentrations of OCPs from both soil depths were in the order of: PF (11.42 ng g^−1^) > BL (1.92 ng g^−1^) > FL (1.58 ng g^−1^) > PGH (1.07 ng g^−1^). The relatively higher level of OCPs in PF might be related to the high affinity of OCPs to soil organic matter content [24]. However, the difference in concentration of OCPs between different land-use types (*p* = 0.069) was not statistically significant.

Samples for this study were collected during the cropping season; thus, the indiscriminate application of lindane and rain washings from surrounding sources might play a big role in escalating the levels OCPs in PF [36]. Likewise, the seasonal application of lindane and rain washings might also have a role in increasing the level of OCPs in FL [53]. A similar phenomenon of OCPs suspension into water bodies and sediments from neighboring sources during summer is reported for East Lake, China [24]. Unlike other land-uses, BL is not expected to obtain OCPs from direct application, however, unsafe storage, disposal, and use of agrochemical might contribute greatly in increasing the OCPs level [5]. Some years ago, the current BL might have been FL, which was expected to receive a considerable amount OCPs. Moreover, BL might receive a considerable quantity OCPs washed from FL, PF and other adjacent point and nonpoint sources.

The concentration of OCPs in PGH was lower than for other land-use types. This is could be due to a limited addition of OCPs from adjacent sources and limited application DDTs and HCHs. The difference in concentration of OCPs among the different land-use types might be linked with the difference in application of pesticides and their degradation [37]. The spreadsheet for all observations acrros the study site from both soil depths is presented in the Appendix A (Table A1). Soil properties can also play a big role in concentration of OCPs in soil. The Pearson’s correlation analysis at (α = 0.01) exhibited a strong positive correlation between individual OCPs and a weak (positive and negative) relationship with selected soil properties. TOC showed a strong correlation value of (R = 0.60) with α-HCH and (R = 0.25) with p,p’-DDT. The relationship among OCPs and soil properties are displayed in the Appendix A (Table A2). Similarly, positive correlations (R = 0.49) with DDTs and (R = 0.52) with HCHs are reported in CauBay River [39], while no correlation between TOC and OCPs is reported for sediments from East Lake [24].

The distributions of HCHs and DDTs across the sampling points in the study area are illustrated in Figure 2 and Figure 3. The ∑HCHs recorded in 0–10 cm from S8, S10, S11, S13 and S18 showed variably higher concentrations than ∑DDTs. Conversely, the ∑DDTs in the 0–10 cm soil layers of S1, S2, S3, S4, S5 and S17 were higher than ∑HCHs. No HCH and DDT were detected at a soil depth of 0–10 cm in S4 and S8. S1, S7, S10, S13, S14, S15, S16 and S18 depicted a higher ∑HCHs than DDTs in 10–20 cm. In contrast, samples from S3, S4, S5 and S17 showed a higher ∑DDTs concentration than ∑HCHs. There was no DDT detected from S7 at a depth of 10–20 cm. The reasons for variation in OCPs is associated with the historical use of technical (DDTs and HCHs), users preferences in using preference chemicals [6]. One-way analysis of variance was conducted to verify the statistical variation in the mean concentrations of OCPs among different land-use types and comparing the difference in concentration between HCHs and DDTs. The results showed that there was a significant among the concentration of OCPs at the α < 0.05 for the four land-use types (F(3,14) = 4.79, *p* = 0.017). However, there were no statistically significant differences in the concentrations of OCPs among sampling sites as determined by one-way ANOVA (F(1,34) = 1.650, *p* = 0.207).

Although there is lack of information about environmental standard ecological risk of values of OCPs for soil in China [24], the Chinese environmental quality standards for soils classified the levels OCPs in soil as: HCHs ≤ 50 and DDTs = 50 ng g^−1^, Grade I; HCHs ≤ 500 and DDT = 500 ng g^−1^, Grade II; and HCH ≤ 1000 and DDT = 1000 ng g^−1^, Grade III [52]. Based on the above classification, obtained concentration < Grade I, Grade I < obtained concentration < Grade II, Grade II < obtained concentration < Grade III are described as negligible, low and moderate pollution levels respectively; whereas obtained concentration > Grade III is classified as a high pollution level [2,44,52]. Accordingly, the levels of OCPs obtained in the study site were much lower than Grade I values for HCHs and DDTs, implying the soil is currently unpolluted.

In addition, the environmental risks of OCPs were evaluated by comparing the obtained values against the Canadian environmental quality standard guideline used by [39]. All the concentrations in this study were less than the interim soil quality guideline (ISQG; 1.42, 3.54, 1.19, 0.94 ng g^−1^) and probable effect level (PEL: 6.75, 8.51, 4.77, 1.38 ng g^−1^) for DDE, DDD, DDT and γ-HCH. The average level of HCHs and DDTs in all samples were lower than the national ocean and atmospheric administration (NOAA) threshold effect levels (TELs) of DDTs in birds (11 ng g^−1^) and soil biological communities (10 ng g^−1^) used in [6]. Generally, the ecological quality values obtained indicated that soils in the study area are suitable for agricultural production without adopting ecological mitigation measures for OCPs [37].

### 3.3. Potential Sources and Composition Organochlorine Pesticides

Computing composition of OCPs is essential in determining the possible sources. Eighty-seven percent of the HCHs concentrations recorded in the current study were β-HCH, while, δ-HCH, α-HCH and γ-HCH contributed 8%, 3% and 2%, respectively. Technical HCHs mainly consisted of four isomers: α-HCH (60–70%), β-HCH (5–12%), γ-HCH (10–15%) and δ-HCH (6–10%) [6,7]. Lindane, which contains 99% γ-HCH, is the other type of HCHs isomer [42]. α-HCH/γ-HCH is the commonly applied HCHs composition ratio to identify the potential sources and degradation of HCHs in environment [42,54]. Even though there is no constant threshold value for α-HCH/γ-HCH to describe its historical use and recent application of technical HCHs [55], several studies use α-HCH/γ-HCH > 3 to indicate application of fresh technical HCHs and values close to zero indicate application of lindane, respectively [39,42,56]. The ratio α-HCH/γ-HCH for all samples except the sample from Bomogang (S12) (α-HCH/γ-HCH = 14.23) were all <3. The results confirmed recent application of lindane as a major and historical use of technical HCHs as a minor source in Huangpi soils. The high proportion of β-HCH also indicated the historical use of technical HCHs [23]. The obtained value was parallel to other reports from soils around industrial parks in Tianjin, China (0.00–3.58) [48] and surface sediments of East lake (0.4–1.07) [24].

Concerning DDTs, technical DDTs contains 75% p,p’-DDT, 15% o,p’-DDT, 5% p,p’-DDE, <0.5%, o,p’-DDE, <0.5%, o,p’-DDD and <5% others unidentified [57]. Other reports stated technical DDTs as a mixture of 85% p,p’-DDT, 15% o,p’-DDT and o,o′-DDT (trace amounts) [58]. The composition of DDTs in this study decreased in order of: p,p’-DDT (72%) > p,p’-DDT (14%) > p,p’-DDE (12%), > p,p’-DDD (3%). p,p’-DDTs have a tendency of converting to p,p’-DDD (under anaerobic) and p,p’-DDE under aerobic conditions. p,p’-DDD/p,p’-DDE and (p,p’-DDE + p,p’-DDD)/∑DDTs ratios are commonly applied to determine the degradation condition and potential sources of DDTs [6,43,49]. p,p’-DDD/p,p’-DDE values < 1 indicate anaerobic degradation of DDTs [6]. In this study, the estimated p,p’-DDD/p,p’-DDE varied from 0.00 to 2.47 with a mean of 0.44. The p,p’-DDD/p,p’-DDE values in 85% of the sampling sites were <1, indicating aerobic degradation of DDTs in the soil. Compared to results of other studies in China, the obtained values were similar to the ratios reported in sediments from East Lake (<1) [24] and in soils from Chinese estuary [6]. However, they were lower than those reported in surface soils of Yellow River (3.15) [59]. The ratio p,p’-DDE + p,p’-DDD/∑DDTs varied between 0.004 (S1) and 1.00 (S10, S11 and S12) with a mean a mean of 0.526, i.e., <1. p,p’-DDE + p,p’-DDD/∑DDTs ratio < 1 implies fresh input of p,p’-DDT [15,40,60], thus indicating application of new technical DDTs. Application of p,p’-DDT for controlling malaria causing mosquito and use of dicofol might be the possible potential sources of DDTs in the study area [15]. The mean ratio of (p,p’-DDE+ p,p’-DDD)/∑DDTs was less than the mean ratio of (0.80) reported from CauBay [39], (0.64) Yellow River estuary [50], (0.66) in sediments from Weihe River China [14]. The potential sources of the investigated OCPs in this study are presented in Table A3.

### 3.4. Human Health Risk Assessment

Hazard quotient measures the noncarcinogenic human health impacts induced from exposure OCPs [36]. The HQ values of OCPs for adults ranged 3.3854 × 10^−8^–3.5405 × 10^−6^ for HCHs and 2.3909 × 10^−6^–1.9130 × 10^−5^ for DDTs, While the HQ range for children were 8.7769 × 10^−8^–9.1791 × 10^−6^ and 2.1669 × 10^−7^–1.7306 × 10^−6^ for HCHs and DDTs, respectively. HQ values of OCPs (DDTs and HCHs) for both adults and children were much lower than the acceptable safe risk level (HQ ≤ 1), implying the obtained OCPs in Huangpi soil are not likely to cause harmful human and environmental health impacts [5]. The obtained HQ results were comparable with that reported from surface water of central China by [15]. The estimated LADD values for children ranged 1.7554 × 10^−7^–1.8358 × 10^−5^ from HCHs and 4.3338 × 10^−7^–3.4612 × 10^−6^ from DDTs, whereas LADD values for adults from HCHs and DDTs were varied 6.7708 × 10^−8^–7.0811 × 10^−6^ and 1.6701 × 10^−7^–1.3400 × 10^−6^, respectively. LADD results suggested that OCPs can pose low risk of cancer for adults. The ∑ILCR (ingestion, dermal contact and inhalation) for HCHs and DDTs for adult age group were 2.2501 × 10^−7^–2.3502 × 10^−5^ and 7.7639 × 10^−7^–6.2007 × 10^−6^, respectively, while the ∑ILCR values for children were varied from 5.8340 × 10^−7^ to 6.1050 × 10^−5^ for HCHs and from 2.0109 × 10^−6^ to 1.6060 × 10^−5^ for DDTs, respectively.

The estimated ∑ILCR values of HCHs and DDTs for both age groups were grouped into very low and low cancer risk level. The LADD, ILCR and HQ values of HCHs were in ascending order of: α-HCH < δ-HCH < γ-HCH < β-HCH. The carcinogenic and noncarcinogenic health impact values of DDTs showed a decreasing trend of: p,p’-DDT > p,p’-DDE > o,p’-DDT > p,p’-DDD. The LADD, ILCR and HQ values of the HCHs and DDTs investigated in this study are presented in Table 4.

The ILCR values of OCPs through inhalation for both age groups were all ≤10^−6^ indicating very low likelihood of cancer through soil inhalation [38]. The cancer risk values OCPs for both age groups through ingestion, dermal contact and inhalation were between very low (negligible) and low cancer risk, implying there is uncertainty in the likelihood of cancer. Generally, the computed average ILCR of OCPs (HCHs and DDTs) of 4.5724 × 10^−6^ for adults and 1.1866 × 10^−5^ for children and the average LADD values of 1.2684 × 10^−6^ for adults and 3.2871 × 10^−6^ for children revealed a low likelihood of cancer risk [61]. According to the estimated ecological and human health risk assessment values, soil in Huangpi is suitable for agricultural production without taking amendment for OCPs.

## 4. Conclusions

This study examined the concentration, distribution, possible source and human health impacts of eight OCPs (HCHs and DDTs) across eighteen sampling sites from four land-use types at the depths of 0–10 cm and 10–20 cm. Results obtained revealed the occurrence of OCPs in the study site. The concentrations of OCPs in soil were higher at 10–20 cm depth than at 0–10 cm soil depth. Concentrations of HCHs in both soil depths were dominated by β-HCH, while p,p’-DDT was the dominant DDTs residue recorded. The average concentrations of OCPs in the four land-use types were higher in PF followed by FL, PGH, and BL, respectively. The concentrations obtained in this study were lower than the threshold concentration of OCPs in soil assigned by Chinese environmental quality standards. The one-way analysis of variance results indicated that there were no statistically significant variations in concentration OCPs among soil depths and sampling points. However, there was a significant difference in the concentration of OCPs in the four land-use types. The Pearson’s correlation coefficient value showed weakly negative and positive correlations between OCPs and selected soil properties. However, there was a strong and moderately positive correlation between individual OCPs. The HCHs isomers and DDTs metabolites ratios used to evaluate the potential sources of OCPs revealed a limited addition of new technical HCHs and fresh application of technical p,p’-DDT is the primary sources DDT in soil. HQ, LADD and ILCR through ingestion, dermal contact, and inhalation suggested an acceptable level of human and environmental health impacts.

## Figures and Tables

**Figure 1 ijerph-16-00146-f001:**
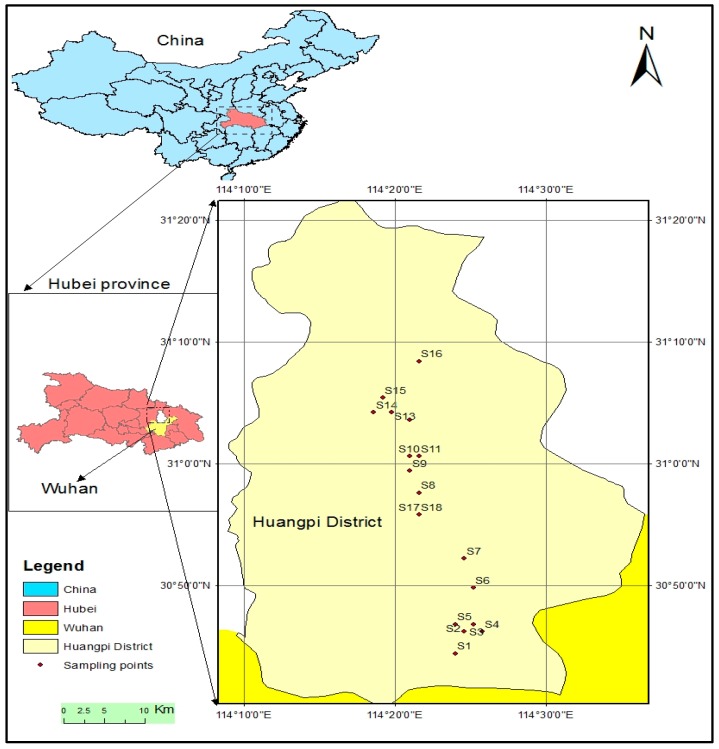
Map of the study area and sampling locations: S1 (Tangjiawan), S2 (Fengdouhu) S3 (Erpaiqu), S4 (Changdi), S5 (Zhujiashan), S6 (Tujiadun), S7 (Zhulinyuan). S8 (Zhoujiawan), S9 (Lishuwan), S10 (Xinyang), S11 (Leqianwan), S12 (Bomogang), S13 (Hanjiafan), S14 (Wanjiatian), S15 (Hongguanshanxiawan), S16 (Dujiatian), S17 (Tianjiaxiaowan), and S18 (Tianjiaxiaowan).

**Figure 2 ijerph-16-00146-f002:**
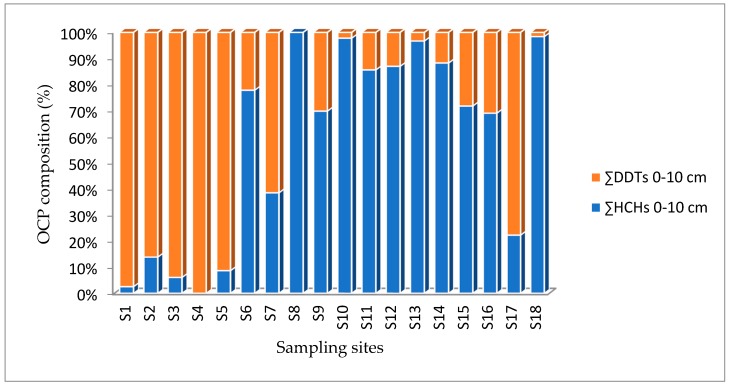
Sum average concentrations organochlorine pesticides (0–10 cm).

**Figure 3 ijerph-16-00146-f003:**
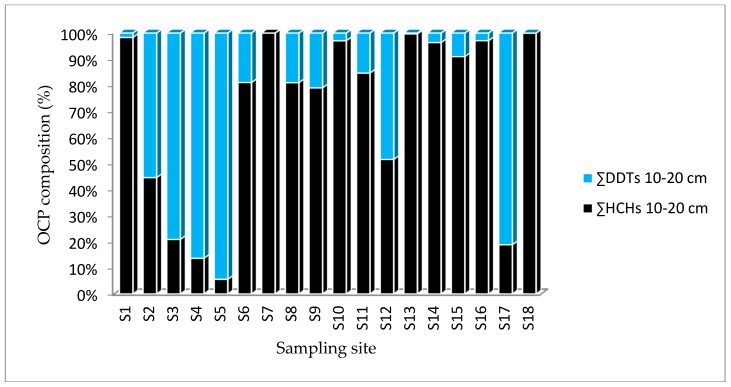
Sum average concentrations organochlorine pesticides (10–20 cm).

**Table 1 ijerph-16-00146-t001:** The frequency, detection limit, quantification limit, standard deviation and average concentration of organochlorine pesticides (ng g^−1^) in Huangpi soils.

OCPs			0–10 cm				10–20 cm			
	DL	QL	Fre	Range	Ave	Std	Fre	Range	Ave	Std
α-HCH	0.05	0.17	94%	ND–0.36	0.07	0.11	100%	0.0002–0.12	0.03	0.04
γ-HCH	0.05	0.16	83%	ND–0.21	0.05	0.07	89%	ND–6.92	0.81	1.91
β-HCH	0.05	0.16	94%	ND–3.80	2.20	4.16	100%	0.01–79.9	7.71	18.62
δ-HCH	0.05	0.16	89%	ND–2.11	0.20	0.49	100%	0.0012–0.18	0.07	0.05
ΣHCHs				ND–16.48	2.52	4.83		0.010–87.12	8.62	20.62
p,p’-DDE	0.05	0.16	89%	ND–0.620	0.19	0.24	89%	ND–1.7	0.34	0.5
o,p’-DDT	0.05	0.15	50%	ND–1.34	0.22	0.48	67%	ND–1.63	0.21	0.47
p,p’-DDD	0.05	0.16	39%	ND–0.76	0.05	0.18	56%	ND–1.77	0.19	0.52
p,p’-DDT	0.05	0.16	61%	ND–13.5	1.17	3.32	56%	ND–8.23	0.7	1.99
ΣDDTs				ND–16.22	1.63	4.22		ND–13.33	1.44	3.48
ΣOCPs				ND–32.7	4.15	9.05		0.01–100.45	10.06	24.10

OCP is organochlorine pesticides, DL is detection limit, QL is quantification limit, HCH is hexachlorocyclohexanes, ND is not detected, Ave is average concentration, Std is standard deviation, Fre is frequency, DDT is dichlorodiphenyltrichloroethane, DDD and DDE are dichlorodiphenyldichloroethylene.

**Table 2 ijerph-16-00146-t002:** Comparison of concentrations of organochlorine pesticides (ng g^−1^) with previous studies.

Study Site/Place	Sample Types	Number of OCP	OCPs	Concentration	Reference
Iran	Sediment	8	DDT and DDE	8.66	[47]
China, Wuhan	Surface water	8	OCP (summer)	5.61–13.62	[15]
			OCP (winter)	3.18–7.73	
Taihu Lake, China	Sediment	8	DDTs	53.9	[13]
			HCHs	1.67	
Xinghua Bay Southeast China	Soil	12	HCHs	3.61 ± 1.76	[40]
			DDTs	8.19 ± 7.28	
Tianjin, China	Soil	8	HCHs	666	[48]
			DDTs	73.9	
CauBay River, Vietnam	Soil	8	HCHs	6.94	[39]
			DDTs	85.24	
Yangtze River, China	Sediment	8	HCHs	1.58	[23]
			DDTs	8.97	
East Central China	Sediments	8	HCHs	2.12 ± 1.08	[49]
			DDTs	11.02 ± 5.11	
Lahore city, Pakistan	Soil	9	HCHs	34.28	[44]
			DDTs	52.56	
Honghu Lake, China	Sediments	8	HCHs	7.72	[24]
			DDTs	9.19	
Yellow River, China	Sediments	10	HCHs	16.45	[50]
			DDTs	2.04	
Pearl River, South China	Paddy soil	8	HCHs	0.00–10.5	[51]
			DDTs	2.23–232	
Beijing, China	Soil	8	HCHs	32	[52]
			DDTs	38	
Huangpi, China	Soil (0–10 cm)	8	HCHs	2.52	This study
			DDTs	1.63	
	Soil (0–10 cm)	8	HCHs	8.62	
			DDTs	1.44	

**Table 3 ijerph-16-00146-t003:** Distribution of OCPs (ng g^−1^) in Huangpi soils across land-use types and two soil depths.

Land-Use Type	Site	0–10 cm			10–20 cm		
		Range	Ave	Std.	Range	Ave	Std.
Farmland (FL)	S3	ND–0.04	0.01	0.01	0.007–1.70	0.61	0.67
	S12	ND–3.00	0.46	1.04	ND–3.11	0.79	1.13
	S15	ND–0.04	0.01	0.01	ND–2.60	0.37	0.90
	S16	0.001–0.03	0.011	0.01	0.001–6.92	0.90	2.43
∑ OCPs FL		0.001–3.11	0.49	1.07	0.008–14.33	2.67	5.13
Pooled Ave ± Std.		1.58 ± 3.10					
Paddy field (PF)	S9	ND–14.00	2.97	4.77	ND–16.30	2.66	5.59
	S10	ND–4.10	0.53	1.44	ND–10.20	1.84	3.64
	S13	ND–12.00	1.64	4.19	0.001–9.76	1.61	3.46
	S14	ND–2.40	0.36	0.83	ND–8.67	1.14	3.04
	S18	ND–0.550	0.08	0.19	ND–79.90	10.01	28.24
∑ OCPs PF		ND–33.05	5.58	11.42	ND–124.83	17.26	43.97
Pooled Ave ± Std.			11.42 ± 27.70				
Plastic greenhouse (PGH)	S4	ND–0.12	0.02	0.04	0.02–1.58	0.30	0.53
	S5	ND–0.78	0.11	0.27	ND–8.23	1.37	2.82
	S6	ND–0.11	0.04	0.04	ND–0.13	0.04	0.05
	S17	ND–1.100	0.24	0.377	0.002–0.15	0.03	0.05
∑ OCPs PGH		ND–2.11	0.41	0.73	0.022–10.09	1.74	3.45
Pooled Ave ± Std.		1.07 ± 2.09					
Barren land (BL)	S1	ND–14.00	1.80	4.93	ND–4.19	0.54	1.47
	S2	0.09–1.30	0.58	0.50	ND–0.13	0.04	0.05
	S7	ND–0.15	0.03	0.05	ND–0.17	0.03	0.06
	S8	ND–0.26	0.09	0.10	ND–0.01	0.002	0.003
	S11	ND–2.40	0.38	0.83	ND–2.18	0.34	0.76
∑ OCPs BL		0.09–18.11	2.88	6.41	ND–6.68	0.95	2.34
Pooled Ave ± Std.		1.92 ± 4.38					
∑ OCPs in 18 samples		0.091–56.38	9.36	19.63	0.023–155.93	22.62	54.89

**Table 4 ijerph-16-00146-t004:** Characterization of carcinogenic and noncarcinogenic health impacts of HCHs and DDTs for adults and children.

Adults						
OCPs	Ingestion	Dermal Contact	Inhalation	∑ILCR	LADD	HQ
α-HCH	1.3542 × 10^−7^	8.9266 × 10^−8^	3.1386 × 10^−10^	2.2501 × 10^−7^	6.7708 × 10^−8^	3.3854 × 10^−8^
γ-HCH	1.2227 × 10^−6^	8.0447 × 10^−7^	2.8340 × 10^−9^	2.0320 × 10^−6^	6.1137 × 10^−7^	3.0568 × 10^−7^
β-HCH	1.4162 × 10^−5^	9.3052 × 10^−6^	3.2824 × 10^−8^	2.3502 × 10^−5^	7.0811 × 10^−6^	3.5405 × 10^−6^
δ-HCH	3.7582 × 10^−7^	2.4831 × 10^−7^	8.7105 × 10^−10^	6.2504 × 10^−7^	1.8791 × 10^−7^	9.3956 × 10^−8^
Ave HCHs	3.9740 × 10^−6^	2.6118 × 10^−6^	9.2107 × 10^−9^	6.5960 × 10^−6^	1.9870 × 10^−6^	9.9350 × 10^−7^
∑HCHs	1.5896 × 10^−5^	1.0447 × 10^−5^	3.6843 × 10^−8^	2.6384 × 10^−5^	7.9481 × 10^−6^	3.9740 × 10^−6^
p,p’-DDE	7.6892 × 10^−7^	1.0150 × 10^−6^	1.7821 × 10^−9^	1.7857 × 10^−6^	3.8402 × 10^−7^	5.4907 × 10^−6^
o,p’-DDT	6.1674 × 10^−7^	8.1409 × 10^−7^	1.4294 × 10^−9^	1.4323 × 10^−6^	3.0800 × 10^−7^	4.4120 × 10^−6^
p,p’-DDD	3.3432 × 10^−7^	4.4130 × 10^−7^	7.7486 × 10^−10^	7.7639 × 10^−7^	1.6701 × 10^−7^	2.3909 × 10^−6^
p,p’-DDT	2.6700 × 10^−6^	3.5245 × 10^−6^	6.1884 × 10^−9^	6.2007 × 10^−6^	1.3400 × 10^−6^	1.9130 × 10^−5^
Ave DDTs	1.0975 × 10^−6^	1.4487 × 10^−6^	2.5437 × 10^−9^	2.5488 × 10^−6^	5.4975 × 10^−7^	7.8559 × 10^−6^
∑DDTs	4.3900 × 10^−6^	5.7949 × 10^−6^	1.0175 × 10^−8^	1.0195 × 10^−5^	2.1990 × 10^−6^	3.1424 × 10^−5^
Ave OCPs	2.5357 × 10^−6^	2.0303 × 10^−6^	5.8772 × 10^−9^	4.5724 × 10^−6^	1.2684 × 10^−6^	4.4247 × 10^−6^
**Children**						
α-HCH	3.5108 × 10^−7^	2.3161 × 10^−7^	3.1386 × 10^−10^	5.8340 × 10^−7^	1.7554 × 10^−7^	8.7769 × 10^−8^
γ-HCH	3.1701 × 10^−6^	2.0971 × 10^−6^	2.8340 × 10^−9^	5.2703 × 10^−6^	1.5850 × 10^−6^	7.9251 × 10^−7^
β-HCH	3.6717 × 10^−5^	2.4250 × 10^−5^	3.2824 × 10^−8^	6.1050 × 10^−5^	1.8358 × 10^−5^	9.1791 × 10^−6^
δ-HCH	9.7436 × 10^−7^	6.4477 × 10^−7^	8.7105 × 10^−10^	1.6201 × 10^−6^	4.8718 × 10^−7^	2.4359 × 10^−7^
Ave HCHs	1.0303 × 10^−5^	6.8059 × 10^−6^	9.2107 × 10^−9^	1.7131 × 10^−5^	5.1514 × 10^−6^	2.5757 × 10^−6^
∑HCHs	4.1213 × 10^−5^	2.7223 × 10^−5^	3.6843 × 10^−8^	6.8524 × 10^−5^	2.0606 × 10^−5^	1.0303 × 10^−5^
p,p’-DDE	1.9935 × 10^−6^	2.6314 × 10^−6^	1.7821 × 10^−9^	4.6249 × 10^−6^	9.9675 × 10^−7^	4.9837 × 10^−7^
o,p’-DDT	1.5989 × 10^−6^	2.1106 × 10^−6^	1.4294 × 10^−9^	3.7096 × 10^−6^	7.9947 × 10^−7^	3.9974 × 10^−7^
p,p’-DDD	8.6675 × 10^−7^	1.1441 × 10^−6^	7.7486 × 10^−9^	2.0109 × 10^−6^	4.3338 × 10^−7^	2.1669 × 10^−7^
p,p’-DDT	6.9223 × 10^−6^	9.1375 × 10^−6^	6.1884 × 10^−9^	1.6060 × 10^−5^	3.4612 × 10^−6^	1.7306 × 10^−6^
Ave DDTs	2.8454 × 10^−6^	3.7559 × 10^−6^	2.5437 × 10^−9^	6.6013 × 10^−6^	1.4227 × 10^−6^	7.1135 × 10^−7^
∑DDTs	1.1381 × 10^−5^	1.5024 × 10^−5^	1.0175 × 10^−8^	2.6405 × 10^−5^	5.6908 × 10^−6^	2.8454 × 10^−6^
Ave OCPs	6.5742 × 10^−6^	5.2809 × 10^−6^	5.8772 × 10^−9^	1.1866 × 10^−5^	3.2871 × 10^−6^	1.6435 × 10^−6^

ILCR is incremental lifetime cancer risks, LADD is lifetime average daily dose and HQ is hazard quotient.

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
