# Peer review of "Determination of Occurrences, Distribution, Health Impacts of Organochlorine Pesticides in Soils of Central China"

_ijerph, 2019, doi:10.3390/ijerph16010146_

Round 1

Reviewer 1 Report

Comments to the manuscript ijerph-406178. Title: "Determination of Occurrences, Distribution, Health Impacts of Organochlorine Pesticides in Soils of One Urban District in China" for the International Journal of Environmental Research and Public Health.

This manuscript is about the analysis of a large family of organohalogenated pesticides (HCHs and DDTs) in soils from Central China. This type of manuscripts centres its interest in the results shown. These pesticides are banned in most of the developed countries since long time ago and in some few cases are used to fight against some plagues are malaria or some few others. However, I have some comments that I below describe.

1.       Previous considerations

Please, revise English in all the text in general. Some sentences need to be revised along the text.

Besides, take care of some technical words:

·         Line 18. ‘Agilent gas chromatograph 7890B equipped with electron capture detector (ECD)’.

·         Line 25. ‘…application new technical p,p’DDT is the main source of DDTs …’

·         Along the complete text, there are many sentences similar to ‘OCP residues from…’ (Line 47), to ‘OCP producer…’ (Line 57) or ‘The mean concentration of HCH residues in Huangpi…’ (Line 209) that need to be especially revised. For instance, lines 245 (OCP levels), 258 (OCP residues), 384 (DDT residues).

·         Line 19 and 130. The common abbreviation for the Electron Capture Detector is ‘ECD’.

·         In my opinion, ‘not detected’ (Nd in the manuscript) would be better to be written as ND.

·         Line 110. In my opinion, Florisil must be written with capital letter for instance, lines 127 or 130).

·         In my opinion, all number must show ‘,’ to identify thousands as for instance in line 45.

·         I think that ‘substracts’ can be an alternative word instead of ‘mediums’ to talk about type of sample.

2.       Materials and methods paragraph.

·         Line 109. Please, add information about quality, company and country for n-hexane. In this case is very special because it is the solvent employed to preparing the stock solution and to rinse glassware before using it.

·         Line 115. About Florisil, is Beijing Yizhong Chemical Plant the supplier company? If not, please, add it.

·         Line 128. When authors talk about activated copper, it is supposed that they activated it, but to what temperature? Did they wash or treated it previously with a reagent to have a better quality?

·         Line 144. Did authors wash glassware only with water? Did they use any kind of soap? These compounds are characterized by high hydrophobicity.

·         Did authors not find interferences at the retention time of HCH isomers? They elute at very short times. Mainly for DDTs, some polychlorinated biphenyl (PCBs) congeners can coelute. How did you solve these potential situations with an ECD? Did you try to use a Mass Spectrometer coupled to a GC system to confirm it?

·         I consider that blanks are essential in any type of analytical study. However, in my opinion, just one blank for every ten samples is few for this study.

·         Section Human Health and Cancer Risk Assessment. Lines 160-161. I do not understand why some values are between parenthesis and some other are not.

·         Please, remove bold type for line 169 and 170. Line 183, add the last ‘t’ to CRinges. In this part of the text, also change RfD into bold type (line 184).

3.       Results

·         Line 197. I do not much agree with ‘detection frequency’ to the number appearing in the Table 1. I consider that a frequency should be a percentage. Maybe it could be more convenient to talk about ‘number of positives’ or something similar.

·         Table 1. Delete italic type to summary of HCHs in the middle of the table.

·         Line 225. Please, revise the decreasing order of DDT isomers for 10-20 cm depth. Among pp’-DDE and pp’-DDD the isomer should be op’-DDT.

·         Line 236. I suppose that ‘OCPs’ is missing after the sum concentration in 10-20 cm layer.

·         Line 236. The sum of the concentrations for Barren Land is higher for 0-10 cm (average 6.41 ng g-1) than for 10-20 cm (0.95 ng g-1). Therefore, that sentence is not completely right.

·         Line 238. According to Table 3, PF land has 5.58 and 14.26 ng g-1 of ΣOCPs for 0-10 cm and 10-20 cm, respectively. Taking into account the differences with those results for FL (0.49 and 2.67), PGH 0.41 and 1.74) and BL (2.88 and 0.95), are they not significantly different?

·         Table 3. In line 236, the low level for 10-20 cm layer is 0.023 ng g-1. Please, as you are using 3 significant digits in the table, write that number in the last raw, 0.023-155.93.

·         The same table 3. Please, change (S9) into bold type in the footnote.

·         Line 273. I am not able to find the Table 6 along the text. Maybe it is table S2 according its information.

·         Line 327. There is mistake when authors talk about the content of DDT technical. In my opinion, according to the reference Kim et al., 2002, the order of percentages should be: pp’-DDT (75%) > op’-DDT (15%) > pp’-DDE (<0.5%) > op’-DDE (<0.5%) > pp’-DDD (<0.5%) > op’-DDD. This composition is referenced from WHO, 1979. However, following documents like ASTDR, 1994 (ATSDR (1994) Toxicological profile for 4,4'-DDT, 4,4'-DDE, and 4, 4'-DDD. Agency for Toxic Substances and Diseases Registry. US Public Health Service. Atlanta, GA.) state that technical DDT is a mixture of three forms, p,p’-DDT (85%), o,p’-DDT (15%), and o,o’-DDT (trace amounts). See also https://www.atsdr.cdc.gov/phs/phs.asp?id=79&tid=20

·         Line 357. According the table 4, LADD HCH values for children range from 4.33E-06 to 3.46E-07.

·         Line 278. Please, revise results, because S3, S4 and S5 show higher DDT concentrations than HCHs in 10-20 cm layer. However, the last sample site S1 shows it for 0-10 cm layer.

Line 281. Authors say that samples from S5, S4, S3 and S17 showed higher DDT concentrations than HCHs. I understand that according the paragraph this must be in 10-20 cm layer. However, S17 belongs to 0-10 cm layer. Please, revise it carefully.

·         Line 283. I understand that it is ‘One-way analysis of variance’. Please, add what is missing.

·         Please, change legend in Y-axis of Figure 2: ‘COP composition’.

·         Table 4. Please, remove bold type for DDT isomers ILCR results. Please, do the same for LADD value in lines 375 and 394.

·         Table S1. I miss data belong to S10-S18 at 10-20 cm depth and their corresponding average. Data in Abstract section (line 20) about β-HCH cannot be confirmed.

·         Table S2. Please, add information about TOC, pH and MC in the footnote.

·         According the introduction section (Line 39), to me is not enough clear if HCHs and DDTs are currently prohibited, because authors state that it is for large-scale production and use. Can this mean that it is still legally producing those OCPs in a minor scale? If I do not understand wrong (Line 75), they are still being used (maybe even produced) and government of Chinas is encouraging their intensive production (Line 78). However, according Grung et al., (2015) (Line 64), OCP production was banned from 1982 (figure 1B). Please, clarify. Is still legal production and use of OCP pesticides in China?

4.       References

·         Please, pay attention to some references:

o    Line 42. Sun et al., 2016b. The list of references does not include it. Only 2016a (#40).

o    Line 91. He et al., 2016. In the list of references is from 2009 (#13). Please, check it.

o    In table 2:

§  Remove ‘G’ for G. Wang et al., 2009.

§  Remove ‘J’ for J. Li et al., 2006.

§  Remove ‘Y’ for Y Shi. Et al., 2005.

o    Line 320. Khureldavaa et al., 2014 is not included in the list of references.

o    Line 341. Remove ‘J’ for J. Zhang et al., 2011.

o    Line 433. According the text, Gao et al., is from 2017.

o    Line 461. Reference 19 does not appear along the text. Please, remove it.

o    Line 51. According the text, the reference 51, Yahaya et al., is from 2017.

o    Is the reference 30 well written? Is Oswer, USEPA (2009) right?

·         Please, carefully follow the instructions for authors to adapt all references: https://www.mdpi.com/journal/ijerph/instructions.

Author 1, A.B.; Author 2, C.D. Title of the article. Abbreviated Journal Name YearVolume, page range. Available online: URL (accessed on Day Month Year).

Author Response

Reviewer 1

·         Please, revise English in all the text in general. Some sentences need to be revised along the text.

Response: It has been revised  

·         Line 18. ‘Agilent gas chromatograph 7890B equipped with electron capture detector (ECD)’.

Response: corrected 

·         Line 25. ‘…application new technical p,p’DDT is the main source of DDTs …’

Response:  Corrected

·         Along the complete text, there are many sentences similar to ‘OCP residues from…’ (Line 47), to ‘OCP producer…’ (Line 57) or ‘The mean concentration of HCH residues in Huangpi…’ (Line 209) that need to be especially revised. For instance, lines 245 (OCP levels), 258 (OCP residues), 384 (DDT residues).

Response: Revision have done according to the comment

·         Line 19 and 130. The common abbreviation for the Electron Capture Detector is ‘ECD’.

Response:  Corrected ECD appeared (Line 18 & 137)

·         In my opinion, ‘not detected’ (Nd in the manuscript) would be better to be written as ND.

Response:  Nd changed by (ND). Presented ((Table 1 & Table 3)

·         Line 110. In my opinion, Florisil must be written with capital letter for instance, lines 127 or 130).

Response: Corrected  

·         In my opinion, all number must show ‘,’ to identify thousands as for instance in line 45.

Response: “,” included appeared in (line 43 & 44)  

·         I think that ‘substracts’ can be an alternative word instead of ‘mediums’ to talk about type of sample.

Response: The word medium has changed to (Environmental components and Sample type)  

·         Line 109. Please, add information about quality, company and country for n-hexane. In this case is very special because it is the solvent employed to preparing the stock solution and to rinse glassware before using it.

Response: The producer company for n-hexane (Sinopharm Chemical Reagent Co. Ltd. Shanghai, China) (Line 112) and for Acetone “Mallinckrodt Baker, Inc.-Phillipsburg, USA” (Line 123)

·          Line 115. About Florisil, is Beijing Yizhong Chemical Plant the supplier company? If not, please, add it.

Response: Supplier company name included (Beijing Yizhong Chemical Plant, Beijing, China) “Line 115”

·         Line 128. When authors talk about activated copper, it is supposed that they activated it, but to what temperature? Did they wash or treated it previously with a reagent to have a better quality?

Response: The copper powder was washed with three times with distilled water and three times with acetone at room temperature. Finally we stored it in acetone.

·         Line 144. Did authors wash glassware only with water? Did they use any kind of soap? These compounds are characterized by high hydrophobicity.

Response: Glassware were washed using distilled water and acetone.

·         Did authors not find interferences at the retention time of HCH isomers? They elute at very short times. Mainly for DDTs, some polychlorinated biphenyl (PCBs) congeners can co-elute. How did you solve these potential situations with an ECD? Did you try to use a Mass Spectrometer coupled to a GC system to confirm it?

Response: The retention time for the compounds was clear raged from 9.00 minutes to 17.02.

·         Section Human Health and Cancer Risk Assessment. Lines 160-161. I do not understand why some values are between parenthesis and some other are not.

Response: Corrected- all the values are in parenthesis

·         Please, remove bold type for line 169 and 170. Line 183, add the last ‘t’ to CRinges. In this part of the text, also change RfD into bold type (line 184). Bold Type removed in (line 169 and 170)

• Table 1. Delete italic type to summary of HCHs in the middle of the table.

Response: Corrected

·         Line 225. Please, revise the decreasing order of DDT isomers for 10-20 cm depth. Among pp’-DDE and pp’-DDD the isomer should be op’-DDT.

Response: Corrected (Line 224)

·         Line 236. I suppose that ‘OCPs’ is missing after the sum concentration in 10-20 cm layer.

Response: it has included (Line 237)

·         Line 236. The sum of the concentrations for Barren Land is higher for 0-10 cm (average 6.41 ng g-1) than for 10-20 cm (0.95 ng g-1). Therefore, that sentence is not completely right.

Response: It has been revised

·         Line 238. According to Table 3, PF land has 5.58 and 14.26 ng g-1 of ΣOCPs for 0-10 cm and 10-20 cm, respectively. Taking into account the differences with those results for FL (0.49 and 2.67), PGH 0.41 and 1.74) and BL (2.88 and 0.95), are they not significantly different?

Response: We considered the average concentration of site in different land-use types. 

   The mean differences were not statistically significant.

·         Table 3. In line 236, the low level for 10-20 cm layer is 0.023 ng g-1. Please, as you are using 3 significant digits in the table, write that number in the last raw, 0.023-155.93.

Response: Corrected: 0.023 –155.93

·         The same table 3. Please, change (S9) into bold type in the footnote

Response: It has been changed

·         Line 273. I am not able to find the Table 6 along the text. Maybe it is table S2 according its information.

Response: It has changed to Table S2 (Line 268)

·         Line 327. There is mistake when authors talk about the content of DDT technical. In my opinion, according to the reference Kim et al., 2002, the order of percentages should be: pp’-DDT (75%) > op’-DDT (15%) > pp’-DDE (<0.5%) > op’-DDE (<0.5%) > pp’-DDD (<0.5%) > op’-DDD. This composition is referenced from WHO, 1979. However, following documents like ASTDR, 1994 (ATSDR (1994) Toxicological profile for 4,4'-DDT, 4,4'-DDE, and 4, 4'-DDD. Agency for Toxic Substances and Diseases Registry. US Public Health Service. Atlanta, GA.) state that technical DDT is a mixture of three forms, p,p’-DDT (85%), o,p’-DDT (15%), and o,o’-DDT (trace amounts). See also https://www.atsdr.cdc.gov/phs/phs.asp?id=79&tid=20

Response:  arrangement corrected according to Kim et al., 2002….  We also considered https://www.atsdr.cdc.gov/phs/phs.asp?id=79&tid=20

·         Line 357. According the table 4, LADD HCH values for children range from 4.33E-06 to 3.46E-07.

Response:  correction made

• Line 278. Please, revise results, because S3, S4 and S5 show higher DDT concentrations than HCHs in 10-20 cm layer. However, the last sample site S1 shows it for 0-10 cm layer.

Response: It has been revised and presented (Line 271-279)

·         Line 281. Authors say that samples from S5, S4, S3 and S17 showed higher DDT concentrations than HCHs. I understand that according the paragraph this must be in 10-20 cm layer. However, S17 belongs to 0-10 cm layer. Please, revise it carefully.

Response: Revised and presented (Line 271-279)

·         Line 283. I understand that it is ‘One-way analysis of variance’. Please, add what is missing.

Response: corrected (line 279)

·         Please, change legend in Y-axis of Figure 2: ‘COP composition’.

Response: Changed to ‘OCP composition’

·         Table 4. Please, remove bold type for DDT isomers ILCR results. Please, do the same for LADD value in lines 375 and 394.

Response:  Bold type has removed

·         Table S1. I miss data belong to S10-S18 at 10-20 cm depth and their corresponding average. Data in Abstract section (line 20) about β-HCH cannot be confirmed.

Response: It was hidden now all the data on Table S1 are visible (Page 13)

·         Table S2. Please, add information about TOC, pH and MC in the footnote.

Response: footnote added (Page 14)

·         According the introduction section (Line 39), to me is not enough clear if HCHs and DDTs are currently prohibited, because authors state that it is for large-scale production and use. Can this mean that it is still legally producing those OCPs in a minor scale? If I do not understand wrong (Line 75), they are still being used (maybe even produced) and government of Chinas is encouraging their intensive production (Line 78). However, according Grung et al., (2015) (Line 64), OCP production was banned from 1982 (figure 1B). Please, clarify. Is still legal production and use of OCP pesticides in China?

Response: There is still production and use of both DDTs and HCHs in different form like Lindane and Dicofol. Moreover there is illegal distribution of these chemicals. The idea is to indicate government of China is encouraging intensive agriculture not intensive use of OCPs.

·         Please, pay attention to some references:

ü  Line 42. Sun et al., 2016b. The list of references does not include it. Only 2016a.

ü  Line 91. He et al., 2016. In the list of references is from 2009 (#13). Please,

ü  In table 2:§ Remove ‘G’ for G. Wang et al., 2009.

ü  Remove ‘J’ for J. Li et al., 2006.

ü  Remove ‘Y’ for Y Shi. Et al., 2005.

ü  Line 320. Khureldavaa et al., 2014 is not included in the list of references.

ü  Line 341. Remove ‘J’ for J. Zhang et al., 2011.

ü  Line 433. According the text, Gao et al., is from 2017.

ü  Line 461. Reference 19 does not appear along the text. Please, remove it.

ü  Line 51. According the text, the reference 51, Yahaya et al., is from 2017.

ü  Is the reference 30 well written? Is Oswer, USEPA (2009) right?

• Please, carefully follow the instructions for authors to adapt all references: https://www.mdpi.com/journal/ijerph/instructions.

 Author 1, A.B.; Author 2, C.D. Title of the article. Abbreviated Journal Name Year, Volume, page range. Available online: URL (accessed on Day Month Year).

Response: All the references have revised according to the author guideline and journal requirement. 

Reviewer 2 Report

Dear Editor, dear Authors, the manuscript “Determination of Occurrences, Distribution, Health Impacts of Organochlorine Pesticides in Soils of One Urban District in China” describes a survey of soil contamination from organochlorine pesticides in a region of central PRC and risk assessment for the populations of adults and children. The topic is within the scope of the Int. J. Environ. Res. Public Health and the overall presentation is worth publication with some minor adjustment. In particular, editing of scientific English and logic may be improved to the benefit of readers and to strengthen the value of the work.

A modification of the title, as “in Soils of central China” would be useful to foreigners for localizing the place. In addition, a suggestion for improvement concerns Figure 1. The scale of the Huangpi district reads as six thousand kilometers, which is as large as the whole of China. However, if it was six kilometers, it looks too small to accommodate so many villages as depicted. This discrepancy puzzles the reader and is worthwhile clarifying.

In section 2.4 Instrumental analysis there is a mixup of topics that should be separated logically in paragraphs. By the way, what is a “Yudian brother furnace”? it looks as a traditional artifact, but it may be not in this context. Again, given that the Authors used HRGC with ECD detection, rather than GC-MS (which is a perfectly acceptable alternative that trades identification for sensitivity), there is a methodological issue with the identification of isomeric pesticides. Since one of the aims of scientific communication is allowing repetition, some data on the robustness of pesticide identification might strengthen the Authors’ point.

In section 2.5, please capitalize and highlight acronyms, in order that readers do not get lost to find the meaning, e.g. Lifetime Average Daily Dose (LADD).

Table 1: LD or DL? QL or LQ?

Figure 2 as such is of little information, even with its comment. Eighteen being the sampling sites, the remaining information is next to obscure. To use 100% as a fixed scale for all sites blinds the differences of absolute contamination among the sites (if the Authors use moles instead of grams they can sum results in their own right). To put together the two depths (1-10 and 10-20 cm) also worsens readability of the interesting results. I dare suggest to make at least two superimposed plots, one above for the upper layer, one below for the lower, and one for the sum, then one for the sum of DDTs and for the sum of HCHs. The results are so important that to present them in a poor way is a real pity and waste of the Authors’ good work. Please put the units in Tables (especially S1: is it still nanograms/gram of soil?): expect that many people will use your data, so please make a correspondence table of S1-S18 to the map in Figure 1.

As for the risk assessment exercise, I know that the software gives the output as reported in Table 4 (risk). However, to give 3-4 significant figures in a large population such as China means that the inverse (1 case over so many people) has a very large uncertainty due to rounding off digits. In addition, the AveOCPs (sum of all sources) is 4E-6, so four times higher than USEPA’s threshold (adults and children taken together). Given the close-to-one-million population of the small investigated area (567 km2, or 20 x 27 km?) this is no small number of extra cases, at least in a Western European perspective. I wonder whether a comment is worthwhile in a PRC perspective, as well. In addition, how were the references of Table 2 selected among the many hundred available?

A few references, such as 12, 38 and 59 cannot be retrieved either from their unreported doi or from google text search. Ref 38 can be reported as a note, 12 and 59 should be fixed or removed.

Best regards

Author Response

Reviewers 2

·         A modification of the title, as “in Soils of central China” would be useful to foreigners for localizing the place. In addition, a suggestion for improvement concerns Figure 1. The scale of the Huangpi district reads as six thousand kilometers, which is as large as the whole of China. However, if it was six kilometers, it looks too small to accommodate so many villages as depicted. This discrepancy puzzles the reader and is worthwhile clarifying.

Response: Thank you for your comments we have thoroughly checked the language problems. We made a title modification, map of the study area (Figure 1) has changed the scale as well has adjusted.

·         In section 2.4 Instrumental analysis there is a mixup of topics that should be separated logically in paragraphs. By the way, what is a “Yudian brother furnace”? it looks as a traditional artifact, but it may be not in this context. Again, given that the Authors used HRGC with ECD detection, rather than GC-MS (which is a perfectly acceptable alternative that trades identification for sensitivity), there is a methodological issue with the identification of isomeric pesticides. Since one of the aims of scientific communication is allowing repetition, some data on the robustness of pesticide identification might strengthen the Authors’ point.

Response: Analysis of OCPs and soil properties has separately presented. The “Yudian brother furnace” has replaced by a muffle furnace (Line 144)

·         In section 2.5, please capitalize and highlight acronyms, in order that readers do not get lost to find the meaning, e.g. Lifetime Average Daily Dose (LADD).

Response: Acronyms capitalized appeared in (Line 160)

·         Table 1: LD or DL? QL or LQ?

Response: Corrected LD & LQ changed by “DL and QL”

·         Figure 2 as such is of little information, even with its comment. Eighteen being the sampling sites, the remaining information is next to obscure. To use 100% as a fixed scale for all sites blinds the differences of absolute contamination among the sites (if the Authors use moles instead of grams they can sum results in their own right). To put together the two depths (1-10 and 10-20 cm) also worsens readability of the interesting results. I dare suggest to make at least two superimposed plots, one above for the upper layer, one below for the lower, and one for the sum, then one for the sum of DDTs and for the sum of HCHs. The results are so important that to present them in a poor way is a real pity and waste of the Authors’ good work. Please put the units in Tables (especially S1: is it still nanograms/gram of soil?): expect that many people will use your data, so please make a correspondence table of S1-S18 to the map in Figure 1.

Response: figure 2 has separated in to two graphs. Figure 2 (0-10 cm) and Figure 3 (10-20c m); Figure 1 has also changed to correspond with table and figures. 

·         As for the risk assessment exercise, I know that the software gives the output as reported in Table 4 (risk). However, to give 3-4 significant figures in a large population such as China means that the inverse (1 case over so many people) has a very large uncertainty due to rounding off digits. In addition, the AveOCPs (sum of all sources) is 4E-6, so four times higher than USEPA’s threshold (adults and children taken together). Given the close-to-one-million population of the small investigated area (567 km2, or 20 x 27 km?) this is no small number of extra cases, at least in a Western European perspective. I wonder whether a comment is worthwhile in a PRC perspective, as well. In addition, how were the references of Table 2 selected among the many hundred available?

Response: Results are presented in four decimal numbers. Concerning to the references of table, we considered the publications on soil and sediments that are used for discussion in the manuscript.

·         A few references, such as 12, 38 and 59 cannot be retrieved either from their unreported doi or from google text search. Ref 38 can be reported as a note, 12 and 59 should be fixed or removed.

Response: Thank you. Corrections made as per the author guideline of the journal.